# Combining Optical Coherence Tomography and Fundus Photography to Improve Glaucoma Screening

**DOI:** 10.3390/diagnostics12051100

**Published:** 2022-04-27

**Authors:** Tomoyuki Watanabe, Yoshimune Hiratsuka, Yoshiyuki Kita, Hiroshi Tamura, Ryo Kawasaki, Tetsuji Yokoyama, Motoko Kawashima, Tadashi Nakano, Masakazu Yamada

**Affiliations:** 1Department of Ophthalmology, The Jikei University School of Medicine, Tokyo 105-8461, Japan; md06-watanabeto@jikei.ac.jp; 2Department of Ophthalmology, Juntendo University School of Medicine, Tokyo 113-8421, Japan; yoshi-h@tkf.att.ne.jp; 3Department of Ophthalmology, Kyorin University School of Medicine, Tokyo 181-8611, Japan; kita@eye-center.org (Y.K.); yamadamasakazu@ks.kyorin-u.ac.jp (M.Y.); 4Department of Ophthalmology and Visual Sciences, Kyoto University Graduate School of Medicine, Kyoto 606-8507, Japan; htamura@kuhp.kyoto-u.ac.jp; 5Center for Innovative Research and Education in Data Science, Institute for Liberal Arts and Sciences, Kyoto University, Kyoto 615-8510, Japan; 6Artificial Intelligence Center for Medical Research and Application, Osaka University Hospital, Suita 565-0871, Japan; ryo.kawasaki@ophthal.med.osaka-u.ac.jp; 7Department of Health Promotion, National Institute of Public Health, Wako 351-0197, Japan; yokoyama.t.aa@niph.go.jp; 8Department of Ophthalmology, Keio University School of Medicine, Tokyo 160-8582, Japan; motoko326@gmail.com

**Keywords:** diagnostic accuracy, glaucoma, optical coherence tomography, ophthalmologist, screening

## Abstract

We aimed to evaluate the accuracy of glaucoma screening using fundus photography combined with optical coherence tomography and determine the agreement between ophthalmologists and ophthalmology residents. We used a comprehensive ophthalmologic examination dataset obtained from 503 cases (1006 eyes). Of the 1006 eyes, 132 had a confirmed glaucoma diagnosis. Overall, 24 doctors, comprising two groups (ophthalmologists and ophthalmology residents, 12 individuals/group), analyzed the data presented in three screening strategies as follows: (1) fundus photography alone, (2) fundus photography + optical coherence tomography, and (3) fundus photography + optical coherence tomography + comprehensive examination. We investigated the diagnostic accuracy (sensitivity and specificity). The respective sensitivity and specificity values for the diagnostic accuracy obtained by 24 doctors, 12 ophthalmologists, and 12 ophthalmology residents were as follows: (1) fundus photography: sensitivity, 55.4%, 55.4%, and 55.4%; specificity, 91.8%, 94.0%, and 89.6%; (2) fundus photography + OCT: sensitivity, 80.0%, 82.3%, and 77.8%; specificity, 91.7%, 92.9%, and 90.6%; and (3) fundus photography + OCT + comprehensive examination: sensitivity 78.4%, 79.8%, and 77.1%; specificity, 92.7%, 94.0%, and 91.3%. The diagnostic accuracy of glaucoma screening significantly increased with optical coherence tomography. Following its addition, ophthalmologists could more effectively improve the diagnostic accuracy than ophthalmology residents. Screening accuracy is improved when optical coherence tomography is added to fundus photography.

## 1. Introduction

Currently, glaucoma is the leading cause of visual impairment in Japan, accounting for 28.6% of cases of recently identified visually impaired individuals [1]. Glaucoma is prevalent in approximately 5% of the Japanese population aged ≥40 years. However, 90% of these cases remain undetected and untreated. Moreover, glaucoma statistics are as follows: normal-tension glaucoma, 3.6%; high-tension primary open-angle glaucoma, 0.3%; primary angle-closure glaucoma, 0.6%; and secondary glaucoma, 0.5% [2,3]. Early diagnosis and treatment are essential to prevent or delay the progression of glaucoma by reducing the intraocular pressure (IOP) [4,5,6,7]. In other words, medical examinations play an important role in the early detection of glaucoma.

The general population undergoes several eye examinations during health checkups, such as a visual acuity test, IOP test, perimetry test, and fundus photography test [8,9,10,11,12,13,14]. The visual acuity test is unlikely to be beneficial as a glaucoma screening method for early detection because glaucoma is unlikely to cause vision loss until it reaches terminal stages. In Japan, normal-tension glaucoma accounts for 72% of all cases. Hence, IOP testing would not be useful. Perimetry is a subjective test that reflects problems, such as fixation losses, false positives, false negatives, and learning effects. Therefore, it may not generate correct test results [15,16,17]. This necessitates identification of the characteristic findings of the optic nerve head and retinal nerve fiber layer (RNFL) defects, using fundus photography as an essential component of glaucoma screening.

However, fundus photography is also associated with some limitations. Factors, such as miosis, corneal opacity, and the opacity of the ocular media, are associated with unclear fundus photographs. In addition, a leopard-spotted fundus is often observed in older adults or a myopic eye, despite a clear image. The condition is common in Asian countries and often makes it difficult to detect an RNFL defect. Furthermore, the optic nerve head findings are evaluated in a two-dimensional manner. Therefore, the inability to perform a three-dimensional evaluation and the experience of the physician interpreting the images may also affect detection efficiency [18,19]. Fundus photographs are widely used as a useful examination for glaucoma detection. Characteristic findings of glaucomatous optic neuropathy (GON), such as focal or diffuse neuroretinal rim thinning, RNFL defects, disc hemorrhage, and beta zone parapapillary atrophy (PPA), can be determined from fundus photographs. However, the interpretation of findings on fundus photographs is subject to variation among examiners because glaucoma must be determined by looking at changes in the color tone and morphology of the RNFL and optic nerve papillary findings on the obtained images [18]. On the other hand, OCT can measure cpRNFL, macular RNFL thickness, ganglion cell layer (GCL) + inner pleciform layer (IPL) thickness, and GCL + IPL + RNFL (GCC: ganglion cell complex) thickness. This enables evaluation of the structure in three dimensions, and quantitative changes can be captured to improve the accuracy of diagnosis [20]. In addition, the recent use of OCT angiography has further improved the accuracy of glaucoma detection by matching the NFLD with a wedge-shaped low-reflective area radiating from the optic nerve papilla [21]. Therefore, OCT can capture the retina as a tomographic image, enabling the understanding of pathological conditions. The combination of fundus photographs and OCT is useful for detecting various diseases, including age-related macular degeneration and Stargardt disease [22,23]. In addition, artificial intelligence is used in various fields, and a high diagnostic accuracy can be expected through continuous learning, which is expected to be built into future health screening systems [24,25].

In 2012, a systematic review on open-angle glaucoma reported the sensitivity and specificity of various screening tests. The sensitivity and specificity of screening with a cup-to-disc ratio of at least 0.59 using fundus photography were 73% and 89%, respectively. In contrast, the sensitivity and specificity of fundus photography for evaluating the RNFL were 75% and 88%, respectively. On using a direct ophthalmoscope and a slit-lamp microscope, the sensitivity and specificity were 60% and 94%, respectively [13]. Furthermore, concerning glaucoma screening using a visual field test, while the sensitivity of a frequency-doubling technology perimeter ranged between 55% and 92%, the specificity ranged between 89% and 100% [26,27,28,29].

There are increasing reports on the accuracy of glaucoma diagnosis using OCT, with some reports demonstrating a sensitivity and specificity of approximately 89% and 95%, respectively [30]. In relation to the diagnosis, spectral domain optical coherence tomography (SD-OCT) is more useful than fundus photography [31,32,33,34,35,36,37,38,39,40,41,42]. In particular, OCT is an excellent auxiliary test for glaucoma diagnosis owing to its non-invasive nature and the rapid generation of accurate tomographic images of the retina and optic nerve head. However, there are currently no clear guidelines for glaucoma diagnosis using OCT, and the results may differ depending on the physicians’ interpretation [43]. In contrast, performing OCT as an additional examination may improve the accuracy of eye examinations in the general population. In other words, glaucoma screening accuracy can be possibly improved by combining fundus photography and OCT rather than using fundus photography alone. Currently, there is no standardized diagnostic procedure for glaucoma using OCT. Furthermore, the diagnostic accuracy improves by the additional examination and varies between ophthalmologists and ophthalmology residents, depending on the examination method [18,19]. Therefore, we aimed to evaluate the accuracy of a screening method that combines both fundus photography and OCT for glaucoma diagnosis among the general population and determine differences in the judgment accuracy between ophthalmologists and ophthalmology residents.

## 2. Materials and Methods

### 2.1. Study Population

Between June 2017 and December 2017, 16 ophthalmology clinics located in three municipalities in Japan (Matsue City, Shimane Prefecture; Sendai City, Miyagi Prefecture; and Setagaya Ward, Tokyo) participated in the study. The target population group included all individuals aged between 40 and 75 years who participated in specific health checkups included in the annual health screening program introduced in 2008 by the Japanese Ministry of Health, Labor, and Welfare. These are the most common medical checkups in Japan. The study methods, participant characteristics, and descriptive statistics have been reported previously [44]. To evaluate the accuracy and feasibility of adult eye examinations, we conducted fundus photography, OCT (circumpapillary RNFL analysis: cpRNFL), SD-OCT (Cirrus HD-OCT [Carl Zeiss Meditec, Oberkochen, Germany]; 3D OCT-2000, 3D OCT-1 [TOPCON, Tokyo, Japan]; RS3000 [NIDEK, Aichi, Japan]; OCT-HS100 [CANON, Tokyo, Japan]; iVue-100 [Optovue, Fremont, California, USA]), Swept Source OCT (SS-OCT) (DRI OCT Triton [TOPCON, Tokyo, Japan]), static perimetry (Humphrey Field Analyzer SITA-Standard 30-2 or 24-2, SITA-Fast 30-2 or 24-2, Humphrey System, Dublin, California, USA), and detailed comprehensive eye examinations (corrected visual acuity, refraction, intraocular pressure [measured using noncontact tonometer], slit-lamp microscopy, and fundus examination) on 1478 participants who visited the ophthalmologic institutions for a fundus examination during their specific health checkups. Among these participants, 1360 successfully underwent all examinations and were included in the original dataset. Of the 118 participants excluded, 39 did not undergo slit-lamp microscopy and fundus examination, 26 had an unknown history of systemic disease and ophthalmology, and 53 agreed to participate in the study but did not undergo the examination. We used a comprehensive ophthalmologic examination dataset obtained from 503 cases (1006 eyes) of 1478 participants who underwent specific health checkups. A definitive diagnosis of glaucoma was established by a central committee comprising three glaucoma specialists using fundus photography, OCT, comprehensive eye examination, and static perimetry. Cases that met the definition criteria for glaucoma, according to the diagnosis by two or more of the three glaucoma specialists, were considered as having glaucoma. Glaucoma was defined as a disease with characteristic changes in the optic nerve head and visual field, and functional and structural abnormalities of the eye. However, it did not include preperimetric glaucoma because of the absence of visual field abnormalities.

The visual field was determined using Anderson’s criteria [45], and the optic disc was evaluated using Foster’s criteria [46]. Glaucoma specialists generally use the temporal-superior-nasal-inferior-temporal map to compare the standard data of cpRNFL thickness with the measured cpRNFL thickness. Cases displaying the disappearance of the double hump pattern, a difference between left and right eye graphs, and an RNFL thickness outside the normal range were used to help diagnose glaucoma [32,33].

### 2.2. Methods

Our accuracy assessment integrated components such as the background information and images (fundus photographs, OCT examinations, and comprehensive eye examinations). Moreover, we used a determination program wherein we presented the clinical information and electronic images of the subjects in an incremental manner. Three different screening methods were used to present the information incrementally. Data were presented using the following three strategies: (1) fundus photography alone (Fds photo), (2) fundus photography with OCT (+OCT), and (3) comprehensive eye examination (+Comp. eye exam.), including corrected visual acuity, refraction, slit-lamp microscopy, IOP, and fundus examination, in addition to fundus photography and OCT. For strategy (3), the ophthalmologic examination findings obtained by the attending physician were displayed. We determined the results for an actual ophthalmologic examination. The determination program was designed such that all cases in strategy (1) would not proceed to strategy (2) until the diagnosis was completed, and that if they proceeded to strategy (2), the results of strategy (1) could not be changed. Similar specifications were used for strategies (2) and (3) (Figure 1). The data were presented such that all evaluators initially studied fundus photography alone, followed by fundus photography + OCT and fundus photography + OCT + comprehensive ophthalmologic examinations. The judges comprised 24 doctors from six different universities (The Jikei University School of Medicine, Kyorin University, Kyoto University, Osaka University, Keio University, and Juntendo University) who were classified into two groups. While one group consisted of 12 ophthalmologists with at least 5 years of experience (ophthalmologist group: clinical experience, 13.9 ± 5.4 years), the other group consisted of 12 ophthalmology residents with <5 years of clinical ophthalmology experience (ophthalmology resident group: clinical experience, 3.0 ± 1.0 years). Each doctor (12 ophthalmologists and 12 ophthalmology residents) examined a similar data set of 503 cases/1006 eyes and determined if each case was normal or required a closer examination (glaucoma or suspected glaucoma).

We examined 24,144 eyes, evaluated by 24 doctors in 1006 eyes of 503 cases, using three diagnostic strategies. These images comprised 3168 glaucomatous eyes.

In addition, we examined 12,072 eyes, diagnosed by 12 ophthalmologists or 12 ophthalmology residents in 1006 eyes of 503 cases, by three diagnostic strategies. The 12,072 images accounted for 1584 glaucomatous eyes.

Three strategies were used to determine whether ophthalmologists and ophthalmology residents could correctly diagnose glaucoma using fundus photographs, OCT, comprehensive ophthalmologic examination, and excluding static perimetry, in cases that had already been diagnosed as normal or glaucoma by glaucoma specialists, using fundus photography, OCT, comprehensive ophthalmologic examination, and static perimetry. The examiner determined focal or diffuse neuroretinal rim thinning, RNFL defects, disc hemorrhage, and beta zone parapapillary atrophy (PPA) from the fundus photographs, and compared cpRNFL thickness from OCT to standard data to determine glaucoma.

Based on the judgment results of each of the three strategies, to determine the accuracy of glaucoma assessment for each of the three screening schemes (Fds photo, +OCT, and +Comp. eye exam.), we calculated the sensitivity, specificity, and their respective 95% confidence intervals; furthermore, we examined differences in the interpretation accuracy between ophthalmologists and ophthalmology residents for glaucoma accuracy. The differences in diagnostic accuracy between the two examinations (Fds photo vs. +OCT; Fds photo vs. +Comp. eye exam.) were analyzed using the McNemar test. A chi-squared test was used to examine the difference in diagnostic accuracy between ophthalmologists and ophthalmology residents. A *p*-value of <0.05 was considered statistically significant. All statistical analyses of diagnostic sensitivity and specificity were performed using EZR [47], a modified version of R commander designed to add frequently used statistical functions in biostatistics [47].

## 3. Results

### 3.1. Patient Demographics

From the 1360 participants, we extracted 503 cases, including glaucoma cases (20%), with clear images that did not affect the diagnosis. Moreover, their information was used as the dataset for accuracy evaluation. Cases with retinal diseases, which were detected using fundus photographs, were excluded from this study (Figure 2). All 503 cases had good image quality and reliable visual fields in both eyes.

The 503 cases included 167 men and 336 women, with an average age of 62.5 (62.5 ± 9.0) years. Ninety-four of the 503 cases and 132 of the 1006 eyes had a confirmed diagnosis of glaucoma, which constituted the diagnostic data set (Table 1) (Figure 3 and Figure 4). Of the 1006 eyes, 764 and 242 eyes underwent SD-OCT and SS-OCT, respectively. The cpRNFL data were analyzed. However, ganglion cell complex parameters were excluded from this study. The image quality criterion for OCT was 30/100 or better.

Among all participants who underwent ophthalmologic examinations during specific health checkups, 503 cases were detected. The fundus photographs were clear, and approximately 20% of these cases were glaucoma.

### 3.2. Comparison of Three Screening Strategies

The number of matched cases diagnosed as glaucoma by glaucoma specialists and glaucoma or suspected glaucoma by 24 doctors were 1754 eyes, 2535 eyes, and 2485 eyes for Fds photo, +OCT, and +Comp. eye exam., respectively (Table 2).

The respective sensitivities, specificities, and 95% confidence intervals of 1006 eyes examined by 24 doctors were as follows: (1) Fds photo: 55.4% (53.6–57.1); 91.8% (91.4–92.2); (2) +OCT: 80.0% (78.6–81.4); 91.7% (91.3–92.1); and (3) +Comp. eye exam.: 78.4% (77.0–79.9); 92.7% (92.3–93.0) (Figure 5 and Figure 6). Compared with Fds photo, +OCT and +Comp. eye exam. had significantly high diagnostic accuracy (McNemar test; *p* < 0.01 and *p* < 0.01, respectively).

The sensitivity for diagnosis by 24 doctors is significantly higher for fundus photography + OCT and fundus photography + OCT + comprehensive exam than for fundus photography alone.

### 3.3. Comparison between Ophthalmologists and Ophthalmology Residents

The number of matched cases diagnosed as glaucoma by glaucoma specialists and glaucoma or suspected glaucoma by 12 ophthalmologists were 877, 1303, and 1264 eyes for Fds photo, +OCT, and +Comp. eye exam., respectively (Table 2).

The respective sensitivities and specificities of the 1006 eyes interpreted by the 12 ophthalmologists were as follows: (1) Fds photo: 55.4% (52.9–57.8); 94.0% (93.5–94.4); (2) +OCT: 82.3% (80.3–84.1); 92.9% (92.3–93.3); and (3) +Comp. eye exam.: 79.8% (77.7–81.8); 94.0% (93.6–94.5) (Figure 7 and Figure 8). Compared with Fds photo, +OCT and +Comp. eye exam. had significantly high diagnostic accuracy (McNemar test; *p* < 0.01 and *p* < 0.01, respectively).

The number of matched cases diagnosed as glaucoma by glaucoma specialists and glaucoma or suspected glaucoma by 12 ophthalmology residents were 877, 1232, and 1221 eyes for Fundus photo, +OCT, and +Comp. eye exam., respectively (Table 2).

The respective sensitivities and specificities of the 1006 eyes interpreted by the 12 ophthalmology residents were as follows: (1) Fds photo: 55.4% (52.9–57.8); 89.6% (89.0–90.2); (2) +OCT: 77.8% (75.6–79.8); 90.6% (90.0–91.1); and (3) +Comp. eye exam.: 77.1% (74.9–79.1); 91.3% (90.8–91.9) (Figure 7 and Figure 8). Compared with Fds photo, +OCT and +Comp. eye exam. had a significantly high diagnostic accuracy (*p* < 0.01 and *p* < 0.01, respectively).

The sensitivity did not significantly differ between ophthalmologists and ophthalmology residents in the Fds photo and +Comp. eye exam. (Chi-square test; *p* = 1.00 and *p* = 0.06, respectively). However, with +OCT, there was a significant difference between ophthalmologists and ophthalmology residents (*p* < 0.01). The specificity significantly differed between ophthalmologists and ophthalmology residents in all three schemes (*p* < 0.01).

## 4. Discussion

We investigated the diagnostic accuracy of glaucoma screening using a combination of fundus photography and OCT in the general population. Our findings demonstrated substantial differences in the diagnostic accuracy between screenings using fundus photography + OCT and fundus photography alone. Since fundus photographs alone cannot quantify the status of glaucomatous optic neuropathy, the addition of OCT would enable the quantification of retinal structures and improve the accuracy of glaucoma diagnosis. The sensitivity of fundus photography alone was 55.4%, which is low for glaucoma screening. Ophthalmologists had the same sensitivity as ophthalmology residents (ophthalmologists: 55.4% vs. ophthalmology residents: 55.4%) in glaucoma screening with higher specificity (ophthalmologists: 94.0% vs. ophthalmology residents: 89.6%), which suggests that the diagnosis was accurate. The fundus photography + OCT method could improve the sensitivity by approximately 25% (Fds photo: 55.4% vs. +OCT: 80.0%). In contrast, no difference was observed in the specificity (Fds photo: 91.8% vs. +OCT: 91.7%).

In a fundus photograph, the retinal and optic nerves are evaluated on a two-dimensional plane. However, OCT displays the results as a three-dimensional image. Therefore, it generates a large amount of information. RNFL defects are particularly useful pieces of information obtained from an OCT [31,34,35,39]. RNFL defects often precede signs, such as optic disc cupping enlargement and visual field defects, and are an early glaucomatous fundus change [48,49]. Thus, OCT may detect glaucoma faster than fundus photography.

The combination of OCT and fundus photography could substantially improve sensitivity. The addition of OCT enabled screening an additional 25% of true glaucoma cases compared with fundus photography alone, thereby suggesting a significant effect. However, the specificity was ≥90% for all screening schemes (Fds photo, +OCT, and +Comp. eye exam.). Screening tests with a high degree of specificity were found extremely useful for positive test results. This can be attributed to the low rate of false positives [50]. Furthermore, there were no significant differences between the specificity of the Fds photo and +OCT. However, we observed a significant but slight difference between those of the +OCT and +Comp. eye exam. Taken together, adding OCT to fundus photography examinations could ensure an accuracy level similar to that achieved by adding comprehensive eye examinations.

Additionally, our findings suggest significant differences in the glaucoma diagnostic accuracy among physicians. Ophthalmologists displayed a significantly higher sensitivity than ophthalmology residents only for the fundus photography + OCT screening scheme. However, ophthalmologists demonstrated a significantly higher specificity than ophthalmology residents in all three screening schemes: moreover, they displayed a 5% improvement in the sensitivity compared with ophthalmology residents on combining OCT with fundus photography. The corresponding specificity was also high (93–94%). Ophthalmologists appeared more capable of diagnosing glaucoma using the fundus photography + OCT screening scheme. However, the quantification of OCT results may pose a challenge by reducing differences in the experiential duration.

Conventionally, ophthalmologists, ophthalmology residents, or optometrists diagnose glaucoma using fundus photography alone. Nonetheless, variations in their ability to accurately diagnose abnormalities in the optic disc using fundus photography pose a problem [18,19]. For evaluating the optic discs of patients with glaucoma, 75 fundus photographs were presented to ophthalmologists (*n* = 6), ophthalmology residents (*n* = 6), and optometrists (*n* = 6), and the sensitivity and specificity for each person were examined. The sensitivity was higher for the ophthalmologists and residents than for optometrists (ophthalmologists: 78%, ophthalmology residents: 78%, and optometrists: 56%). The specificity was 60%, 47%, and 53% for ophthalmologists, ophthalmology residents, and optometrists, respectively. Therefore, the specificity values were significantly higher for ophthalmologists [18]. Thus, our findings are supported by those of previous studies [18,19].

This study has several strengths. First, the target population comprised community-dwelling adults. Ninety percent of patients with potential glaucoma were unaware of their condition and were not undergoing any treatment [2]. Instead of examining outpatients already diagnosed with glaucoma, we were able to conduct our study under conditions closely similar to that of a real ophthalmological examination using an actual population-based data set involving patients undergoing specific health checkups. Second, we independently evaluated three screening methods. The first method was fundus photography alone, which is the conventionally used diagnostic method. The second method involved adding OCT to fundus photography. The third method included the addition of data on visual acuity, refraction, and IOP, as well as data collected from slit-lamp microscopy and fundus tests for fundus photography and OCT; it was similar to a comprehensive eye examination. This enabled comparing the accuracy of the aforementioned three diagnostic methods.

Nevertheless, our study has some limitations. First, certain components of the study, such as the OCT model and perimetry program, were not completely uniform. However, the consideration of factors, such as the actual examination settings and clinical practice for OCT models, renders it impossible to use a similar model in every facility. Therefore, we could conduct our evaluations in a realistic situation. The use of various OCT models was one of the limiting factors. Despite some reports of no differences in the diagnostic accuracy between SD-OCT and SS-OCT [51,52], a single model is preferable. In this study, we used data from a multicenter study, which included various OCT models. While 19 out of the 503 cases in the perimetry program were examined using the Humphrey–Fast program, the remaining cases were examined using the Humphrey Field Analyzer SITA-Standard program. A re-examination of all but 19 cases generated identical findings. Therefore, the aforementioned limitation does not invalidate our results. Subsequently, the presentation method was another limitation. Regarding the +Comp. eye exam., we envisioned the type of evaluation to be conducted during an actual ophthalmology examination. In addition to fundus photography and OCT, we added data corresponding to a comprehensive eye examination, i.e., one obtained during the physician’s examination (corrected visual acuity, refraction, IOP, slit-lamp microscopy, and fundus examination). However, the situation was different from an actual medical examination and could not be considered exactly similar. In particular, the amount of information obtained from the actual examinations was greater. Therefore, the sensitivity and specificity of screening strategy (3) might have been underestimated. Third, considering their association with RNFL thinning, RNFL abnormalities may help diagnose diseases such as myopia, Alzheimer’s disease, Parkinson’s disease, multiple sclerosis, schizophrenia, sleep apnea syndrome, obesity, metabolic syndrome, and anemia. Therefore, there lies the possibility of false positivity [53,54,55,56,57,58,59,60,61,62,63,64,65]. Fourth, the study was limited to the following areas: Matsue, Shimane; Sendai, Miyagi; and Setagaya, Tokyo. Despite the need to carefully extrapolate these results to external municipalities, the abovementioned areas comprised patients who underwent specific health checkups and were eligible for eye examinations at a medical institution. Therefore, these results may reflect those of the general population. Fifth, we did not consider the cost implications. From a public health perspective, cost-effectiveness is one of the 10 factors to be assessed while screening [66]. Despite adding the comprehensive examination to fundus photography + OCT screening, the difference in the sensitivity was minimal. Considering the screening cost, adding OCT to fundus photography appeared sufficient. This necessitates the determination of the cost-effectiveness of adding OCT to conventional eye examinations in the future.

## 5. Conclusions

Glaucoma screening by combining OCT with fundus photography showed a 25% higher sensitivity than using fundus photography alone. However, there were no significant differences in the specificity. Therefore, fundus photography and OCT appears more effective than fundus photography alone for glaucoma screening. The sensitivity and specificity values of the diagnoses by ophthalmologists were significantly higher than those by ophthalmology residents. Therefore, ophthalmologists may more effectively conduct glaucoma screening than ophthalmology residents using fundus photography and OCT.

## Figures and Tables

**Figure 1 diagnostics-12-01100-f001:**
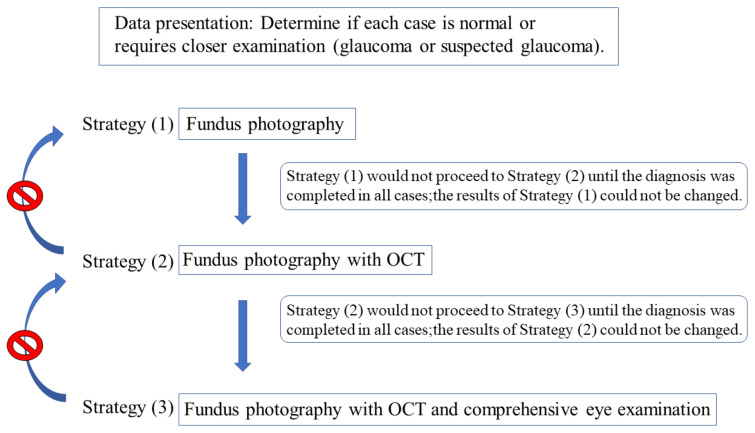
Diagnostic method based on three strategies. Diagnosis was performed in the order of the strategies (1), (2), and (3). Once the judges proceeded to the next strategy, their answers could not be changed.

**Figure 2 diagnostics-12-01100-f002:**
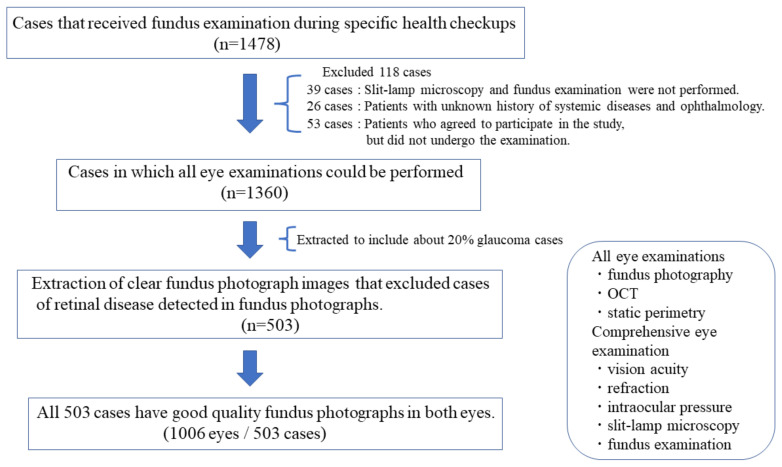
Criteria for extracting 503 cases.

**Figure 3 diagnostics-12-01100-f003:**
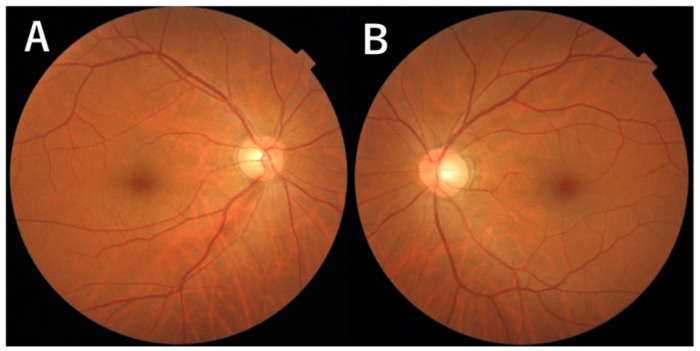
Photographs of the fundus. (**A**): Photograph of the fundus shows a defect in the nerve fiber layer located in the inferotemporal area in the right eye. Enlarged view of the optic disc at the 6 o’clock disc edge. (**B**): No abnormal findings in the left eye.

**Figure 4 diagnostics-12-01100-f004:**
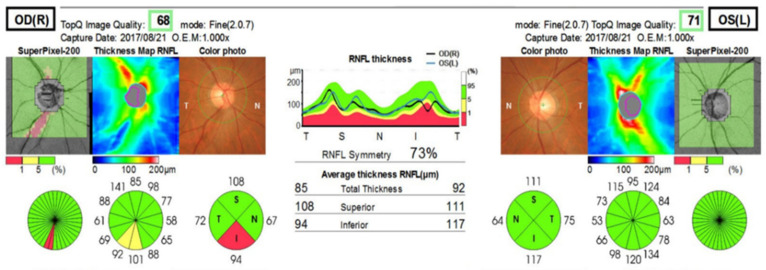
RNFL thickness: Swept Source OCT (Triton [TOPCON]). OCT indicates thinning of the nerve fiber layer in the right eye.

**Figure 5 diagnostics-12-01100-f005:**
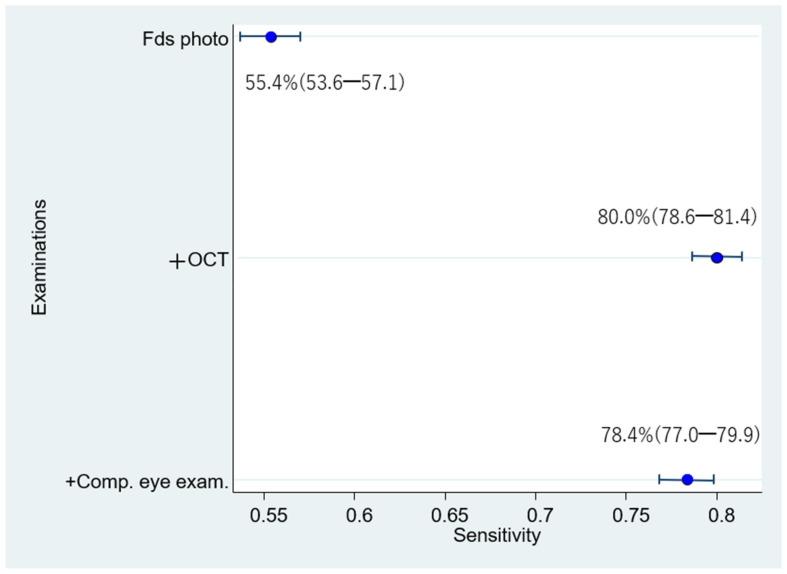
Sensitivity for diagnosis by 24 doctors.

**Figure 6 diagnostics-12-01100-f006:**
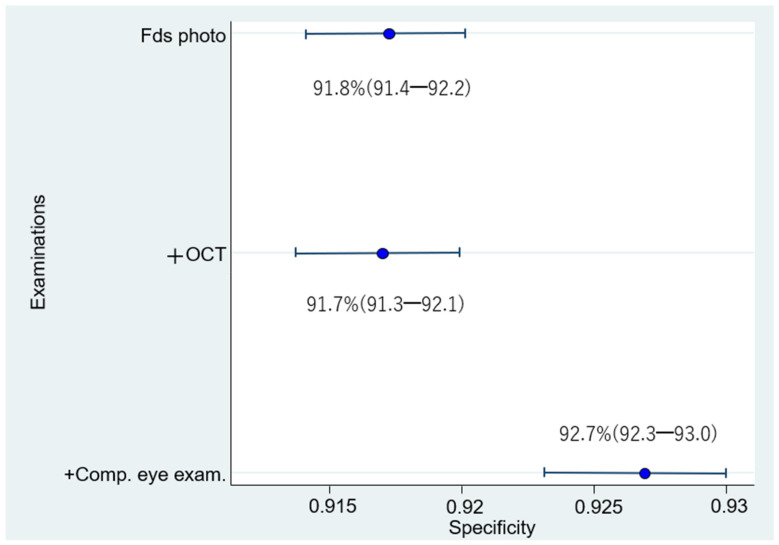
Specificity for diagnosis by 24 doctors. The specificity for diagnosis by 24 doctors is significantly higher for fundus photography + OCT + comprehensive examination than for fundus photography alone or fundus photography + OCT.

**Figure 7 diagnostics-12-01100-f007:**
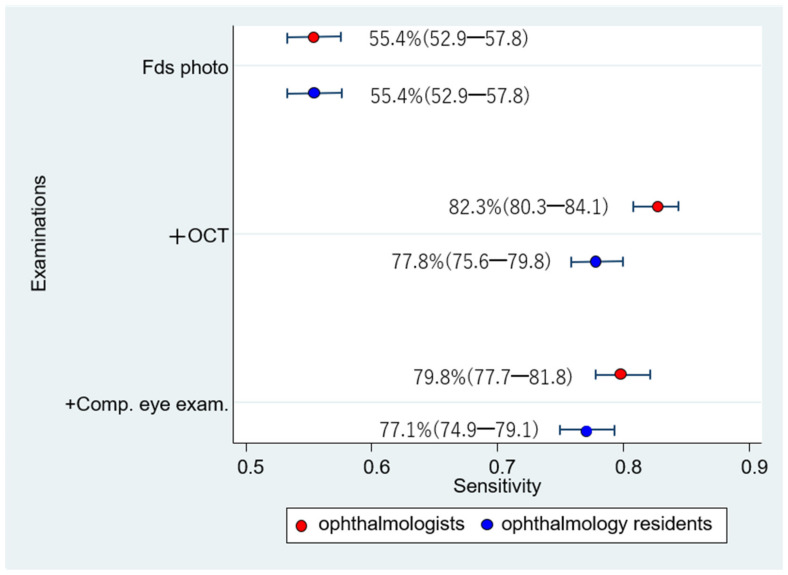
Sensitivity for diagnosis by ophthalmologists and ophthalmology residents. Comparing the sensitivity between ophthalmologists and ophthalmology residents, the sensitivity of ophthalmologists is significantly higher for fundus photography + OCT than that of ophthalmology residents.

**Figure 8 diagnostics-12-01100-f008:**
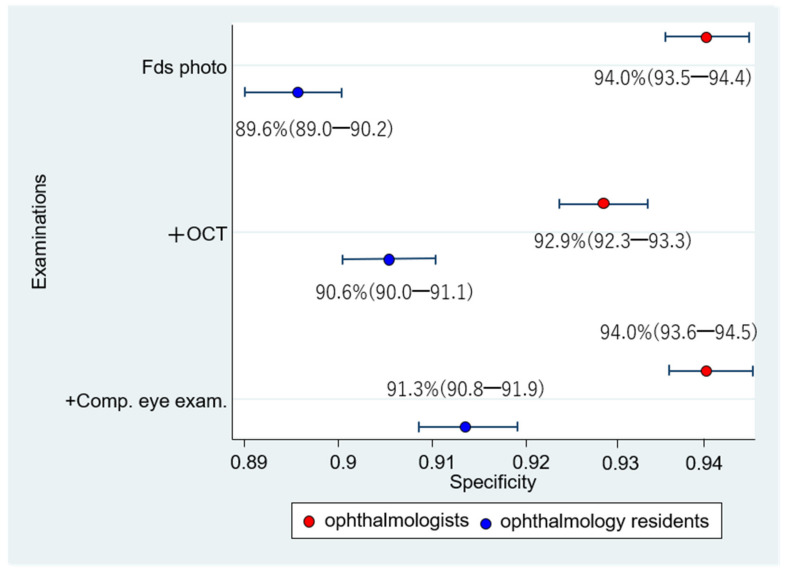
Specificity for diagnosis by ophthalmologists and ophthalmology residents. Comparing the specificity between the ophthalmologists and ophthalmology residents, the specificity of ophthalmologists is significantly higher for fundus photography alone, fundus photography + OCT, and fundus photography + OCT + comprehensive exam than that for ophthalmology residents.

**Table 1 diagnostics-12-01100-t001:** Characteristics of study participants.

Demographic Characteristics				
Age, Years	62.5 ± 9.0			
Sex (Male/Female), *n*/*n*	167/336			
Ocular Characteristics	All (*n* = 1006)	Normal (*n* = 874)	Glaucoma (*n* = 132)	*p*-Value †
Spherical equivalent, D	−1.12 ± 2.58	−0.96 ± 2.46	−2.12 ± 3.11	<0.01
logMAR	−0.04 ± 0.08	−0.05 ± 0.07	−0.01 ± 0.11	<0.01
IOP, mmHg	14.6 ± 2.8	14.6 ± 2.8	14.9 ± 2.9	0.18
MD (dB)	−0.76 ± 5.50	−0.13 ± 1.81	−4.96 ± 13.79	<0.01
PSD (dB)	2.42 ± 2.12	1.97 ± 1.26	5.41 ± 3.69	<0.01
Average thickness cpRNFL (μm)	95.9 ± 13.1	98.4 ± 11.5	79.4 ± 10.9	<0.01

*n* = number; D = diopter; logMAR = logarithm of the minimum angle of resolution; IOP = intraocular pressure; MD = mean deviation; PSD = pattern standard deviation; and cpRNFL = circumpapillary retinal nerve fiber layer. † Mann–Whitney U test.

**Table 2 diagnostics-12-01100-t002:** Number of diagnosed eyes that required a closer examination (glaucoma or suspected glaucoma) in the three diagnostic strategies/Number of eyes diagnosed with glaucoma.

	Fundus Photography	Fundus Photography with OCT	Fundus Photography with OCT and Comprehensive Eye Examination
24 doctors	1754/3168	2535/3168	2485/3168
12 ophthalmologists	877/1584	1303/1584	1264/1584
12 ophthalmology residents	877/1584	1232/1584	1221/1584

OCT, optical coherence tomography.

## Data Availability

Not applicable.

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
