# Peer review of "Combining Optical Coherence Tomography and Fundus Photography to Improve Glaucoma Screening"

_diagnostics, 2022, doi:10.3390/diagnostics12051100_

Round 1

Reviewer 1 Report

Authors have performed a comparative study to evaluate the accuracy of clinical screening of glaucoma using multiple retinal imaging modalities (fundus photography and optical coherence tomography).  The diagnostic accuracy of glaucoma is determined in terms of sensitivity and specificity. They also compared the evaluation performed by ophthalmologists and ophthalmology residents. They concluded that the addition of OCT imaging could significantly improve the diagnostic accuracy of glaucoma screening. Furthermore, ophthalmologists have performed a better glaucoma screening than ophthalmology residents. The manuscript   requires a major revision based on the following comments:

  1. In the introduction authors have briefly narrated the fundus photography (FP) and OCT. The most important difference between FP and OCT is that FP can provide only two dimensional en face image (topography) of the retinal fundus. Whereas, OCT can provide depth-resolved 2D and 3D images of retina. Additionally, OCT angiography (OCTA) can image retinal blood flow (blood vessels), which is quite helpful for the assessment of RFNL. Authors should provide more details of the imaging capabilities of OCT and FP.
  1. Authors should provide a brief description of OCT and fundus photography techniques and their wide application scenarios. For example, a combined OCT and fundus imaging system could be used for the detection of multiple disease including AMD, Stargardt disease etc. Authors may use the following  references to highlight these approaches(https://doi.org/10.1038/s41598-021-95320-z, https://doi.org/10.1117/1.JBO.24.6.066011)
  2. How did you define the sensitivity and specificity of each imaging technique and assessment performed by ophthalmologists and ophthalmology residents. It is not mentioned that how did authors calculated the sensitivity and specificity. Need to clearly detail.
  3. It is not explained that what are the different retina parameters/biomarkers they used to study the glaucoma
  4. How did authors measured the average thickness of the RFNL in Table 1.
  5. Image quality of Figure 1 and Figure 2 are very poor. 
  6. Authors need to provide a representative image of OCT and FP for glaucoma and non-glaucoma eyes they examined. Need to show the variation of RFNL thickness (OCT images) in both cases.
  7. The usage of “fundus photography+OCT” in conclusion is inappropriate
  8. Line 88 is the repetition of line 83.

Author Response

Our point-by-point response to all comments and suggestions is below:

We are grateful to the reviewers for taking the time and effort necessary to review our manuscript and provide us with these valuable comments and suggestions. We have revised our manuscript accordingly. Please note that our changes to the manuscript are indicated by text in red-colored font.

Responses to the comments by reviewer #1.

Reviewer’s comment

In the introduction authors have briefly narrated the fundus photography (FP) and OCT. The most important difference between FP and OCT is that FP can provide only two dimensional en face image (topography) of the retinal fundus. Whereas, OCT can provide depth-resolved 2D and 3D images of retina. Additionally, OCT angiography (OCTA) can image retinal blood flow (blood vessels), which is quite helpful for the assessment of RFNL. Authors should provide more details of the imaging capabilities of OCT and FP.

Response to Reviewer

We thank the reviewer for the comment. We have added to the introduction details about fundus photography and OCT imaging capabilities.

Changes in the Introduction

Introduction (Page: 2, Line: 67):

Fundus photographs are widely used as a useful examination for glaucoma detection. Characteristic findings of glaucomatous optic neuropathy (GON), such as focal or diffuse neuroretinal rim thinning, RNFL defects, disc hemorrhage, and beta zone parapapillary atrophy (PPA), can be determined in fundus photographs. However, the interpretation of findings on fundus photographs is subject to variation among examiners because glaucoma must be determined by looking at changes in the color tone and morphology of the RNFL and optic nerve papillary findings on the obtained images [18]. On the other hand, OCT can measure cpRNFL, macular RNFL thickness, ganglion cell layer (GCL)+inner pleciform layer (IPL) thickness, and GCL+IPL+RNFL (GCC: ganglion cell complex) thickness. This enables evaluation of the structure in three dimensions, and quantitative changes can be captured to improve the accuracy of diagnosis [20]. In addition, the recent use of OCT angiography has further improved the accuracy of glaucoma detection by matching the NFLD with a wedge-shaped low-reflective area radiating from the optic nerve papilla [21].

Reviewer’s comment

Authors should provide a brief description of OCT and fundus photography techniques and their wide application scenarios. For example, a combined OCT and fundus imaging system could be used for the detection of multiple disease including AMD, Stargardt disease etc. Authors may use the following  references to highlight these approaches(https://doi.org/10.1038/s41598-021-95320-z, https://doi.org/10.1117/1.JBO.24.6.066011)

Response to Reviewer

We thank the reviewer for the suggestion. We have cited "In vivo multimodal retinal imaging of disease-related pigmentary changes in retinal pigment epithelium" and "Directional optical coherence tomography reveals melanin concentration-dependent scattering properties of retinal pigment epithelium" as references. We have added an explanatory statement that the combination of OCT and fundus photography can provide a useful diagnosis.

Changes in the Introduction

Introduction (Page: 2, Line: 80):

Therefore, OCT can capture the retina as a tomographic image, enabling the understanding of pathological conditions. The combination of fundus photographs and OCT is useful for detecting various diseases, including age-related macular degeneration and Stargardt disease [22,23].

Reviewer’s comment

How did you define the sensitivity and specificity of each imaging technique and assessment performed by ophthalmologists and ophthalmology residents. It is not mentioned that how did authors calculated the sensitivity and specificity. Need to clearly detail.

Response to Reviewer

We thank the reviewer for the important point to clarify the significance of this study.

Three strategies were used to determine whether ophthalmologists and ophthalmology residents could correctly diagnose glaucoma using fundus photographs, OCT, comprehensive ophthalmologic examination and excluding static perimetry, in cases that had already been diagnosed as normal or glaucoma by glaucoma specialists, using fundus photography, OCT, comprehensive ophthalmologic examination, and static perimetry.

Accordingly, we have revised parts of the manuscript as follows.

Changes in the Materials and Methods

Materials and Methods (Page: 4, Line: 186):

Three strategies were used to determine whether ophthalmologists and ophthalmology residents could correctly diagnose glaucoma using fundus photographs, OCT, comprehensive ophthalmologic examination and excluding static perimetry, in cases that had already been diagnosed as normal or glaucoma by glaucoma specialists, using fundus photography, OCT, comprehensive ophthalmologic examination, and static perimetry.

Reviewer’s comment

It is not explained that what are the different retina parameters/biomarkers they used to study the glaucoma.

Response to Reviewer

We thank the reviewer for the comment.

Characteristic findings of glaucomatous optic neuropathy (GON) were determined from fundus photographs, and cpRNFL was compared with the normal data on OCT.

Accordingly, we have revised a part of the manuscript as follows.

Changes in the Materials and Methods

Materials and Methods (Page: 4, Line: 191):

The examiner determined focal or diffuse neuroretinal rim thinning, RNFL defects, disc hemorrhage, and beta zone parapapillary atrophy (PPA) from the fundus photographs, and compared cpRNFL thickness from OCT to standard data to determine glaucoma.

Reviewer’s comment

How did authors measured the average thickness of the RFNL in Table 1.

Response to Reviewer

We thank the reviewer for the comment.

Average thickness cpRNFL was measured in 503 cases/1,006 eyes with either SD-OCT (Cirrus HD-OCT, 3D OCT-2000, 3D OCT-1, RS3000, OCT-HS100, iVue-100) or SS-OCT (DRI OCT Triton) models.

Reviewer’s comment

Image quality of Figure 1 and Figure 2 are very poor.

Response to Reviewer

We thank the reviewer for bringing this to our attention.

The images in Figures 1 and 2 have been adjusted, proved quality.

Changes in Figures 1 and 2

Figure 1 (Page: 5, Line: 210):

Figure 2 (Page: 6, Line: 231):

New Figure 1 and 2

Reviewer’s comment

Authors need to provide a representative image of OCT and FP for glaucoma and non-glaucoma eyes they examined. Need to show the variation of RFNL thickness (OCT images) in both cases.

Response to Reviewer

We thank the reviewer for the thoughtful suggestion.

We have presented representative glaucomatous and non-glaucomatous eye cases as new figures in the revised manuscript.

Changes in Figures 3 and 4

Changes in the Results

Figure 3 (Page: 7, Line: 245):

Figure 4 (Page: 7, Line: 249)

Results (Page: 5, Line: 220)

Reviewer’s comment

The usage of “fundus photography+OCT” in conclusion is inappropriate.

Response to Reviewer       

We thank the reviewer for bringing this to our attention.

We have changed "fundus photography+OCT" to "fundus photography and OCT" in the Conclusion.

Changes in the Conclusion

Conclusion (Page: 12, Line: 414, 418):

fundus photography and OCT

Reviewer’s comment

Line 88 is the repetition of line 83.

Response to Reviewer

We thank the reviewer for the comment.         

We have deleted " Currently, there is no standardized diagnostic procedure for glaucoma using OCT " in line 88 of the old manuscript.

Reviewer 2 Report

Glaucoma screening by combining OCT with fundus photography showed a 25% higher sensitivity than using fundus photography alone. However, there were no significant differences in the specificity. Therefore, fundus photography+OCT appears more effective than fundus photography alone for glaucoma screening. The sensitivity and specificity values of the diagnoses by ophthalmologists were significantly higher than those by ophthalmology residents. Therefore, ophthalmologists may more effectively conduct glaucoma screening than ophthalmology residents using fundus photography+OCT. The paper is organized well. However, there are some points to be considered.

  1. The model parameters are not fully presented, which makes the reproduction difficult.
  2. Words in Figures are too small. Please enlarge the small words.
  3. Please introduce some representative studies of novel approach of artificial general intelligence in the Introduction Section, including: Robust Spike-Based Continual Meta-Learning Improved by Restricted Minimum Error Entropy Criterion; SAM: A Unified Self-Adaptive Multicompartmental Spiking Neuron Model for Learning with Working Memory.
  4. Please discuss the hardware implementation of the proposed model. Some researches can be discussed in terms of this point, including: CerebelluMorphic: large-scale neuromorphic model and architecture for supervised motor learning; Scalable digital neuromorphic architecture for large-scale biophysically meaningful neural network with multi-compartment neurons; Neuromorphic context-dependent learning framework with fault-tolerant spike routing.
  5. The proposed algorithm is not well justified in the paper, which needs more theoretical analyses. I cannot understand why this method can improve the performance.
  6. Grammar is expected to be further improved.

Author Response

Our point-by-point response to all comments and suggestions is below:

We are grateful to the reviewers for taking the time and effort necessary to review our manuscript and provide us with these valuable comments and suggestions. We have revised our manuscript accordingly. Please note that our changes to the manuscript are indicated by text in red-colored font.

Responses to the comments by the reviewer #2.

Reviewer’s comment

 The model parameters are not fully presented, which makes the reproduction difficult.

Response to Reviewer

We thank the reviewer for the comment.

The examiner detected glaucomatous findings by determining focal or diffuse neuroretinal rim thinning, RNFL defects, disc hemorrhage, and beta zone parapapillary atrophy (PPA) from fundus photographs. In the OCT results, cpRNFL thickness was compared with standard data to determine glaucoma. We have added the following to the manuscript.

Changes in the Materials and Methods

Materials and Methods (Page: 4, Line: 191):

The examiner determined focal or diffuse neuroretinal rim thinning, RNFL defects, disc hemorrhage, and beta zone parapapillary atrophy (PPA) from the fundus photographs, and compared cpRNFL thickness from OCT to standard data to determine glaucoma.

Reviewer’s comment

Words in Figures are too small. Please enlarge the small words.

Response to Reviewer

We thank the reviewer for bringing this to our attention.

We have increased the font size of the text in the figures.

Reviewer’s comment

Please introduce some representative studies of novel approach of artificial general intelligence in the Introduction Section, including: Robust Spike-Based Continual Meta-Learning Improved by Restricted Minimum Error Entropy Criterion; SAM: A Unified Self-Adaptive Multicompartmental Spiking Neuron Model for Learning with Working Memory.

Response to Reviewer

We thank the reviewer for the suggestion.

We have cited "Robust Spike-Based Continual Meta-Learning Improved by Restricted Minimum Error Entropy Criterion" as a reference in the Introduction section.

Changes in the Introduction

Introduction (Page: 2, Line: 84):

In addition, artificial intelligence is used in various fields, and high diagnostic accuracy can be expected through continuous learning, which is expected to be built into future health screening systems [24,25].

Reviewer’s comment

 Please discuss the hardware implementation of the proposed model. Some researches can be discussed in terms of this point, including: CerebelluMorphic: large-scale neuromorphic model and architecture for supervised motor learning; Scalable digital neuromorphic architecture for large-scale biophysically meaningful neural network with multi-compartment neurons; Neuromorphic context-dependent learning framework with fault-tolerant spike routing.

Response to Reviewer

We thank the reviewer for the suggestion.

We have cited "Large-Scale Neuromorphic Model and Architecture for Supervised Motor Learning" as a reference in the Introduction section.

Changes in the Introduction

Introduction (Page: 2, Line: 84):

In addition, artificial intelligence is used in various fields, and high diagnostic accuracy can be expected through continuous learning, which is expected to be built into future health screening systems [24,25].

Reviewer’s comment

 The proposed algorithm is not well justified in the paper, which needs more theoretical analyses. I cannot understand why this method can improve the performance.

Response to Reviewer

We thank the reviewer for the comment.

Since fundus photographs alone cannot quantify the status of glaucomatous optic neuropathy, the addition of OCT would enable the quantification of retinal structures and improve the accuracy of glaucoma diagnosis.

We have added this to our Discussion.

Changes in the Discussion

Discussion (Page: 11, Line: 315):

Since fundus photographs alone cannot quantify the status of glaucomatous optic neuropathy, the addition of OCT would enable the quantification of retinal structures and improve the accuracy of glaucoma diagnosis.

Reviewer’s comment

Grammar is expected to be further improved.

Response to Reviewer

We thank the reviewer for the suggestion. We have checked the grammar.

Round 2

Reviewer 1 Report

Authors have addressed all the questions satisfactorily. No further questions or concerns.

Reviewer 2 Report

I am satisfied with the reivision. No comments remained.